# Metal Surface Defect Detection Method Based on TE01 Mode Microwave

**DOI:** 10.3390/s22134848

**Published:** 2022-06-27

**Authors:** Meng Shi, Lijian Yang, Songwei Gao, Guoqing Wang

**Affiliations:** School of Information Science and Engineering, Shenyang University of Technology, Shenyang 110870, China; yanglijian888@163.com (L.Y.); gaosongwei888@163.com (S.G.); wangguoqing@sut.edu.cn (G.W.)

**Keywords:** TE01 mode, microwave reflection method, field distribution equation, reflection coefficient, return loss, crack detection

## Abstract

With the aim of addressing the difficulty of detecting metal surface cracks and corrosion defects in complex environments, we propose a detection method for metal surface cracks and corrosion defects based on TE01-mode microwave. The microwave detection equations of cracks and corrosion defects were established by the Maxwell equations when the TE01 mode was excited by microwaves, and the relationship model between the defect size and the microwave characteristic quantity was established. A finite integral simulation model was established to analyze the influence of defects on the microwave electric field, magnetic field, and tube wall current in the rectangular waveguide, as well as the return loss at the defect; an experimental platform for the detection of metal surface cracks and corrosion defects was built. The absolute value of the return loss of the microwave reflected wave increased, and with the increase of the defect width, the microwave detection frequency at the defect decreased. The TE01-mode microwave has good detection ability for metal surface cracks and corrosion defects and can effectively detect cracks with a width of 0.3 mm.

## 1. Introduction

Metal materials support the main structure of social and economic life, such as nuclear power plants, long-distance oil and gas pipelines, large mechanical equipment, and storage tanks. With the increase in the service life of metal materials, the structural properties of metal materials change, and surface cracks and corrosion of metal materials can lead to catastrophic accidents. As a new type of non-destructive testing method with high efficiency and sensitivity, microwave can prevent the occurrence of catastrophic accidents; effectively reduce national economic losses; and ensure the safety of national life, ecological environment, and social stability.

Magnetic flux leakage detection technology is sensitive to volume defects of ferromagnetic materials, and magnetic flux leakage detection technology can only detect ferromagnetic metal materials [1,2]. Piezoelectric ultrasonic detection technology can detect metal and non-metallic multilayer materials, such as pipeline anti-corrosion layers, which require a couplant and have high requirements for the detection environment [3]. Electromagnetic ultrasonic testing technology can detect material crack damage, and the attenuation characteristics of ultrasonic waves in materials are complex [4,5]. In terms of metal surface defect detection, eddy current detection technology is sensitive to near-surface defects in conductive materials, and the signal-to-noise ratio of eddy current coils is low at low frequencies [6,7,8]. Laser inspection has the advantage of non-contact, all-view inspection [9,10]. The image processing method has a good detection effect on surface crack detection [11,12], and it has higher requirements for the image acquisition method and image quality. Extensive research has been conducted in the field of microwave metal detection. Wu B et al. conducted research based on the SAR imaging algorithm of compressive sensing reconstruction to detect foreign debris in multilayer circular dielectric structures [13]. JU Y et al. established a microwave resonance model, analyzed the microwave signal in the time domain, and evaluated the thinning of the pipeline [14,15]. Katagiri T et al. analyzed the TEM-mode/TE01-mode conversion, and the microwave reflection was related to the transmission frequency, the size of the transmission line, the electromagnetic properties, and the roughness of the material surface [16,17,18]. Chen G et al. designed two TEM–TM01 and TEM–TM02 incident probes to obtain better propagation mode purity and propagation directionality, detect defects in different orientations, and analyze the influence of propagation mode conversion at bends on detection signals [19,20,21,22]. YU YT et al. established a finite integral numerical model to analyze the relationship between the reflection coefficient phase and the defect edge position [23]. Experts and scholars have analyzed the relationship between microwave intrinsic impedance, defect position, and wavelength, but they have not specifically analyzed the relationship between microwave defect signals and defects. This paper proposes a relationship model between return loss, reflection coefficient, and defect size that can be used to qualitatively analyze the relationship between defect signals and defects. It has guiding significance for microwave detection of large steel surface defects.

Compared with traditional non-destructive testing methods, microwave testing, as a new non-destructive testing section, has the characteristics of non-contact, no coupling agent, and low energy loss in media such as oil and gas. With the aim of addressing the problem of microwave detection of metal surface defects, this paper built a TE01-mode microwave detection system for metal surface cracks and corrosion defects, established a microwave detection model for cracks and corrosion defects, and established defect size and microwave reflected wave characteristic parameter models. The TE01-mode microwave simulation model at the defect was established to analyze the distribution of electric field, magnetic field, and pipe wall current at the defect, and the energy loss of the echo loss at the defect under the simulation model was obtained. In order to verify TE01’s ability to detect different types of metal surface defects, a TE01-mode microwave metal surface defect microwave detection experimental platform was built. The TE01-mode microwave has high sensitivity for detecting metal surface defects.

## 2. Metal Surface Defect Identification Method

The metal surface defects were detected by TE01 mode, and the microwave detection frequency was 5.2–6 GHz. The metal surface defects were detected by the rectangular waveguide probe, using the microwave reflection method, and the metal surface defect detection model was established.

### 2.1. Transverse Wave (TEmn Wave) in Rectangular Waveguide

The rectangular waveguide probe establishes a rectangular coordinate model, and the time–harmonic field in the rectangular waveguide probe propagates along the *z*-axis. The passive region Maxwell equation is as follows:(1)$$\nabla \times \vec{E}=-j\omega \mu \vec{H}$$
(2)$$\nabla \times \vec{H}=j\omega \varepsilon \vec{E}$$
where $\vec{E}$ and $\vec{H}$ are the electric and magnetic fields in the rectangular waveguide, respectively; ω is the angular frequency; ε is the permittivity; and μ is the permeability. Taking the curl at both ends of the above equation, the Helmholtz equation can be used to obtain the longitudinal component equation in the rectangular waveguide, and the transverse component equation in the rectangular waveguide is expressed in the longitudinal direction. The TE wave field component in the rectangular waveguide is as follows:(3)$$E_{x}=j\frac{\omega \mu }{k_{c}^{2}}H_{0}\frac{n\pi }{b}\cos\left(\frac{m\pi }{a}x\right)\sin\left(\frac{n\pi }{b}y\right)e^{j\beta z}$$
(4)$$E_{y}=-j\frac{\omega \mu }{k_{c}^{2}}H_{0}\frac{m\pi }{a}\sin\left(\frac{m\pi }{a}x\right)\cos\left(\frac{n\pi }{b}y\right)e^{j\beta z}$$
(5)$$E_{z}=0$$
(6)$$H_{x}=\frac{j\beta }{k_{c}^{2}}H_{0}\frac{m\pi }{a}\sin\left(\frac{m\pi }{a}x\right)\cos\left(\frac{n\pi }{b}y\right)e^{j\beta z}$$
(7)$$H_{y}=\frac{j\beta }{k_{c}^{2}}H_{0}\frac{n\pi }{b}\cos\left(\frac{m\pi }{a}x\right)\sin\left(\frac{n\pi }{b}y\right)e^{j\beta z}$$
(8)$$H_{z}=H_{0}\cos\left(\frac{m\pi }{a}x\right)\cos\left(\frac{n\pi }{b}y\right)e^{j\beta z}$$

When the TE01 mode propagates in a rectangular waveguide, the cutoff frequency is 5.2 GHz. The field distribution is shown in Equations (3)–(8) for when the TE01 mode propagates in a rectangular waveguide, where m is 0, and n is 1. At this time, the electric field in the *y*-direction is 0, the electric field in the *z*-direction is 0, and the magnetic field in the *x*-direction is 0. At this time, when the TE01 mode propagates, the loss in the rectangular waveguide wall is small, and the propagation mode is more stable. $H_{0}$ is the amplitude constant of the magnetic field in the rectangular waveguide. Because the time-harmonic field propagates along the *z*-axis direction, the components of the above vector equation are expanded in three directions, β is the propagation constant, and $k_{c}$ is the transverse cutoff wave number, where we have the following:(9)$$k^{2}=\omega^{2}\mu \varepsilon$$
(10)$$\beta =\sqrt{k^{2}-k_{c}^{2}}$$
(11)$$k_{c}=\sqrt{{\left(\frac{m}{a}\right)}^{2}+{\left(\frac{n}{b}\right)}^{2}}$$

In the above formula, *a* and *b* are the length and width of the rectangular waveguide, respectively; *a* > *b*; *m* represents the number of half-waves of the TE wave along the *x*-direction; and n represents the number of half-waves of the TE wave along the *y*-direction. It can be seen from Equations (3)–(8) that, when the propagation mode in the rectangular waveguide probe is the TE01 mode, only Ex, Hy, and Hz propagate in the rectangular waveguide change, showing a standing wave distribution along the *b*-direction.

The tube wall current in the rectangular waveguide was determined by the tangential magnetic field, and the tube wall current at the crack was as follows:(12)$$J=\hat{n}\times H_{\tan}$$

When *y* = *b* and *y* = 0 at the broad side of the rectangular waveguide, the wall current is as follows:(13)$$J|{}_{y=0}=H_{0}\cos\left(\frac{\pi }{b}y\right)e^{-j\omega z}$$
(14)$$J|{}_{y=b}=H_{0}\cos\left(\frac{\pi }{b}y\right)e^{-j\beta z}$$

When *x* = 0 and *x* = *a* at the long side of the defect, the wall current is as follows:(15)$$J|{}_{x=0}=H_{0}\cos\left(\frac{\pi }{a}y\right)e^{-j\beta z}-\frac{j\beta }{k_{c}^{2}}H_{0}\frac{\pi }{a}\sin\left(\frac{\pi }{a}y\right)e^{-j\beta z}$$
(16)$$J|{}_{x=a}=\frac{j\beta }{k_{c}^{2}}H_{0}\frac{\pi }{a}\sin\left(\frac{\pi }{a}y\right)e^{-j\beta z}-H_{0}\cos\left(\frac{\pi }{a}y\right)e^{-j\beta z}$$

At *y* = 0 and *y* = *b*, the TE10 mode has only the *x*-direction component, which is equal in size and in the same direction. At *x* = 0 and *x* = *a*, it is composed of *y*- and *z*-direction components. Therefore, the wall current only propagates on the rectangular waveguide wall. When a defect occurs on the metal surface, the wall current at the defect propagates along the defect, and the energy is lost at the defect.

### 2.2. Microwave Detection Model of Metal Surface Defects

When the microwave is incident vertically, the metal surface has no defects, and the electric field and magnetic field of the reflected wave are not distorted. When defects are present on the metal surface, the reflected electric field and magnetic field have energy loss at the defect. The microwave reflection coefficient is the ratio of the microwave reflected wave’s electric field to the microwave incident wave’s electric field; the microwave incident wave’s electric field is $E_{i}$, the microwave reflected wave’s electric field is $E_{r}$, and the microwave reflection coefficient of the rectangular defect is as follows:(17)$$\Gamma =\frac{E_{r}}{E_{i}}$$

#### 2.2.1. Metal Surface Defect Detection Model in Cartesian Coordinates

When the metal surface defect is a crack defect, the length of the defect is *l*_1_, the width of the defect is *l*_2_, and the depth of the defect is *h*. The schematic diagram of the rectangular defect on the metal surface is shown in Figure 1.

When the microwave propagates in the TE01 mode in the rectangular waveguide probe, there is only an electric field in the *x*-direction and a magnetic field in the *y*- and *z*-direction; the field distribution in the rectangular waveguide probe can be seen from Equations (3)–(8). According to Equation (12), when the metal surface defect is rectangular, the reflection coefficient is as follows:(18)$$\Gamma_{1}=\frac{k_{c}^{2}\left[\begin{array}{cc} 0 & -\mu_{1}\\ \mu_{1} & 0 \end{array}\right]\left[\begin{array}{c} \frac{-m\pi }{l_{1}}H_{10}\sin\left(\frac{m\pi }{l_{1}}x\right)\cos\left(\frac{n\pi }{l_{2}}y\right)\\ \frac{-n\pi }{l_{2}}H_{10}\cos\left(\frac{m\pi }{l_{1}}x\right)\sin\left(\frac{n\pi }{l_{2}}y\right) \end{array}\right]e^{j\beta Z+h}}{k_{1c}^{2}\mu H_{0}\frac{\pi }{b}\sin\left(\frac{\pi }{b}y\right)e^{j\beta Z}}$$
where $\mu_{1}$ is the magnetic permeability at the defect, $k_{1c}^{2}$ is the cutoff wave number of the derived distortion field at the defect, $H_{10}$ is the peak value of the magnetic field at the defect, $m_{1}$ is the number of half-waves in the *x*-direction of the derived distortion field at the defect, $n_{1}$ is the *y*-direction of the derived distortion field at the defect The number of half waves of, $l_{1}$ is the defect length, and $l_{2}$ is the defect width. It can be seen from Equation (18) that, when there are defects on the metal surface, the microwave reflection coefficient decreases with the increase of the crack depth on the metal surface.

#### 2.2.2. The Metal Surface Defect Detection Model under Cylindrical Coordinates

When the metal surface defect is corrosion-shaped, the coordinates of any point in the defect are (*r*, *φ*, *z*), and the microwave propagates along the *z*-axis direction. The schematic diagram of the metal surface corrosion defect is shown in Figure 2.

When the defect is a cylindrical defect, the radius of the defect is R, the angle of the defect is ϕ, the depth of the defect is *h*, and the field distribution equation of the TE01-mode microwave at the defect is as follows:(19)$${\vec{E}}_{r}=\pm j\frac{\omega \mu m}{k_{c}^{2}\vec{r}}H_{0}J_{m}\left(\frac{u_{mn}}{R}\vec{r}\right)\left(\begin{array}{l} \sin m\varphi \\ \cos m\varphi \end{array}\right)e^{-j\beta z}$$
(20)$${\vec{E}}_{\varphi }=j\frac{\omega \mu }{k_{c}}H_{0}J_{m}^{\prime }\left(\frac{u_{mn}}{R}\vec{r}\right)\left(\begin{array}{l} \cos m\varphi \\ \sin m\varphi \end{array}\right)e^{-j\beta z}$$
(21)$$E_{z}=0$$
(22)$${\vec{H}}_{r}=-j\frac{\beta }{k_{c}}H_{0}J_{m}^{\prime }\left(\frac{u_{mn}}{R}\vec{r}\right)\left(\begin{array}{l} \cos m\varphi \\ \sin m\varphi \end{array}\right)e^{-j\beta z}$$
(23)$$H_{\varphi }=\pm H_{0}\frac{j\beta }{k_{c}^{2}\vec{r}}J_{m}\left(\frac{u_{mn}}{R}\vec{r}\right)\left(\begin{array}{l} \sin m\varphi \\ \cos m\varphi \end{array}\right)e^{-j\beta z}$$
(24)$${\vec{H}}_{z}=H_{0}J_{m}\left(\frac{u_{mn}}{R}\vec{r}\right)\left(\begin{array}{l} \cos m\varphi \\ \sin m\varphi \end{array}\right)e^{-j\beta z}$$
where *m* represents the wave number of microwave propagation distributed along the circumference, *n* represents the number of fields distributed along the radius, ω is the angular frequency, μ is the permeability, $k_{c}$ is the cutoff wavenumber, and β is the propagation constant. Moreover, $H_{0}$ is the peak value of the magnetic field during the propagation of the TEmn mode. It can be observed from the above formula that the TE wave in the pipeline propagates sinusoidally along the axial direction. $J_{m}$ is the m-order Bessel function of the first kind, $J_{m}^{\prime }$ is the derivative of the m-order Bessel function of the first kind, and *u_mn_* is the TE01 modulus, the root of the Bessel function. Corrosion defects have only a Hz magnetic field near the tube wall, and the tube wall current is calculated as follows:(25)$$J|{}_{r=R}=H_{0}\cos\left(\frac{\pi }{l_{2}}y\right)e^{-j\beta h}$$

Due to the defect of the waveguide, the current cutting of the tube wall causes energy loss. Due to the defect, the geometric size of the waveguide probe is discontinuous, and the current cutting of the tube wall leads to changes in the return reflection coefficient and return loss.

From the cylindrical defect field distribution equation, it can be known that, when cylindrical defects exist on the metal surface, the microwave reflection coefficient at this time is as follows:(26)$$\Gamma_{1}=\frac{k_{c}^{2}\left[\begin{array}{cc} 0 & -\mu_{1}\\ \mu_{1} & 0 \end{array}\right]\left[\begin{array}{c} H_{10}k_{1c}J_{m}^{\prime }\left(\frac{u_{mn}}{R+h}\right)\left(\frac{\cos m\varphi }{\sin m\varphi }\right)\left(e^{j\beta z+h}\right)\\ \pm \frac{1}{r}\mu_{1}H_{10}J_{m}\left(\frac{u_{mn}}{R+h}\right)\left(\frac{\sin m\varphi }{\cos m\varphi }\right)\left(e^{j\beta z+h}\right) \end{array}\right]}{k_{1c}^{2}\mu H_{0}\frac{\pi }{b}\sin\left(\frac{\pi }{b}y\right)e^{j\beta Z}}$$
where $\mu_{1}$ is the magnetic permeability at the defect, $k_{1c}^{2}$ is the cutoff wave number of the derived distortion field at the defect, $H_{10}$ is the peak value of the magnetic field at the defect, R is the radius of the defect, and h is the depth of the defect. It can be seen that, when the metal surface defects are cracks and corrosion defects, the reflection coefficient is related to the depth of the defect, and the reflection coefficient decreases with an increase in the defect size.

## 3. Simulation Model of Microwave Defect Detection in TE01 Mode

We built a microwave detection model for metal surface defects; the size of the rectangular waveguide probe is 59 mm × 29 mm, and the microwave incident wave frequency is 5.2–6 GHz to establish a microwave detection model for crack defects and corrosion defects. During the propagation of the microwaves in the waveguide, they propagate in the form of electric field, magnetic field, and wall current in the waveguide. When there is a defect on the surface of the tested metal, the boundary conditions at the defect change, resulting in the distortion of the field distribution in the waveguide and jumps in the propagation mode.

### 3.1. TE01-Mode Microwave Defect Detection Model

#### 3.1.1. TE01-Mode Microwave Defect Detection Model at Crack Defects

We established the microwave simulation model of different defects, crack defects (50 mm × 0.3 mm × 5 mm), and defects in different types of rectangular coordinates: triangular-prism defects (20 mm × 10 mm × 5 mm), trapezoid defects (upper surface, 20 mm × 10 mm; lower surface, 20 mm × 4 mm; and depth, 5 mm).

As shown in Figure 3, when the metal surface is free of defects, the electric field in the rectangular waveguide is distributed in two complete cycles, the phase difference between the magnetic field and the electric field is 90°, and the tube wall current propagates along the tube wall along the tangential direction of the magnetic field. When there is no defect, there is no energy loss in the rectangular waveguide, the wall current is related to the tangential component of the magnetic field, and the wall current propagates along the surface of the rectangular waveguide. When there are defects on the metal surface, the defects hinder the propagation of the wall current, the energy is lost, and the condition of energy loss is detected.

As shown in Figure 4, when the metal surface defect is a crack defect, the microwave propagation period in the rectangular waveguide is similar to that without defect, the 0.3 mm defect does not change the propagation period in the rectangular waveguide, and part of the energy leaks at the defect. The crack surface of the defect cuts the tube wall current and hinders the propagation of the tube wall current. The distribution of the microwave field propagating in the rectangular waveguide is correspondingly distorted; the boundary conditions at the defect are changed; and the peak values of the electric field, the magnetic field, and the tube wall current in the rectangular waveguide increase. In the reflection coefficient formula, as H0 increases, the reflection coefficient decreases.

As shown in Figure 5, when the metal surface defect is a triangular-prism defect, the defect opening in the rectangular waveguide is 10 mm wide. In the rectangular waveguide, the phase of the microwave propagation period changes significantly, and the defect causes the microwave field distribution at the defect to be distorted; moreover, the peak value of the electric field, the magnetic field, and the tube wall current increase.

As shown in Figure 6, when the metal surface defect is a trapezoidal defect, the upper surface of the trapezoidal defect is the same as the triangular prism defect, and the phase distribution in the rectangular waveguide is similar. When the current is distorted, the peak values of the electric field, the magnetic field, and the tube wall current propagating in the rectangular waveguide increase. In the reflection coefficient formula, as H0 increases, the reflection coefficient decreases.

It can be seen from Figure 3, Figure 4, Figure 5 and Figure 6 that, with the increase of the defect width on the metal surface, the field distribution in the rectangular waveguide is destroyed, the phase of the propagation period changes, and the field distribution in the rectangular waveguide is distorted. It can be seen from Figure 5 and Figure 6 that the triangular prism and the trapezoid have the same defect surface size and the same phase of the field distribution change. The peak field distribution at the defects is listed in Table 1.

As shown in Table 1, when there is no defect in the pipe, the magnetic field in the rectangular waveguide is equal to the pipe wall current value, the pipe wall current is equal to the tangential magnetic field at the pipe wall, and the pipe wall current is equal to the magnetic field. As defects appear on the metal surface, part of the energy of the electric field, magnetic field, and tube wall current leaks to the defect, resulting in an increase in the peak value of the electric field, magnetic field, and tube wall current in the waveguide. When the metal surface defect is a 0.3 mm–wide crack, the energy leakage is relatively high. As the defect width increases, the magnitude of each field component increases. When the defect surface size is the same, the larger the defect volume, the greater the energy leakage, and the larger the energy peak in the rectangular waveguide.

#### 3.1.2. TE01-Mode Microwave Defect Detection Model at Corrosion Defects

A defect simulation model under the cylindrical coordinate system was established, and the defect sizes are cylindrical defect (φ10 mm × 5 mm), cone defect (φ10 mm × 5 mm), hemispherical defect (φ10 mm), and semi-cylindrical defect (section is 20 mm × 5 mm × φ 10 mm).

As shown in Figure 7, when the metal surface defect is a cone defect, the upper surface of the cone defect is a cone defect (Φ1 mm), the tube wall current propagates along the arc in the rectangular waveguide, and part of the tube wall current propagates along the surface of the defect, without forming an obstacle trend of the current propagation in the tube wall. This causes the phase change of the microwave propagation cycle in the rectangular waveguide to be small, which, in turn, causes the distortion of the microwave field distribution at the defect, and the peak values of the electric field, the magnetic field, and the tube wall current in the rectangular waveguide increase. In the reflection coefficient formula, as H0 increases, the reflection coefficient decreases.

As shown in Figure 8, when the metal surface defect is a hemispherical defect, the upper surface of the hemispherical defect is consistent with the cone defect, and the midfield distribution of the hemispherical defect is similar to the cone defect; the inner wall of the hemispherical defect is spherical, and the tangential direction of the inner surface of the sphere changes constantly, hindering the current propagates through the tube wall; the electric field, magnetic field, and tube wall current are significantly distorted at the defect; and the energy loss is greater than that of the cone defect, reducing the reflection coefficient.

As shown in Figure 9, when the metal surface defect is a cylindrical defect, the upper surface of the cylindrical defect is the same as that of the cone defect, the field distribution is similar to that of the cone defect, and a new interface is formed on the lower surface of the cylindrical defect that hinders the wall current propagation; the electric field, magnetic field, and tube wall current at the defect are significantly distorted; the peak value of the electric field, magnetic field, and tube wall current propagating in the rectangular waveguide increases; H0 in the reflection coefficient increases; and the reflection coefficient decreases.

As shown in Figure 10, when the metal surface defect is a semi-cylindrical defect, the upper surface of the defect is consistent with the triangular prism, and the phase change in the rectangular waveguide is similar to a triangular prism; the inner wall of the semi-cylindrical defect is arc-shaped, and at the upper and lower bases of the semi-cylindrical, a new separation surface is formed to hinder the wall current the propagation to the greatest extent. The electric field, the magnetic field, and the wall current are distorted the most at the defect. The reflection coefficient is reduced the most, and the energy loss is the most.

The tube wall current propagates along the arc direction for cone defects and hemispherical and cylindrical defects. At this time, the tangent direction of the magnetic field changes with the arc tangent release, and the defect weakens the cutting degree of the tube wall current; moreover, the ability to detect the arc-shaped defect on the upper surface is weakened, the semi-cylindrical defect is rectangular on the metal surface, and the inner surface of the defect is arc-shaped. The wall current is related to the tangential direction of the magnetic field. When the inner wall of the defect is arc-shaped, the current direction of the tube wall changes continuously; thus, the TE01-mode microwave has the best detection ability for semi-cylindrical defects. The peak value of cylindrical defect field distribution is listed in Table 2.

As shown in Table 2, the electric field, magnetic field, and tube wall current energy distributions are lower at the defects with circular metal surface shapes, such as cones, hemispheres, and cylinders, thus reducing the energy loss in the rectangular waveguide; after the cone defect tube wall current enters the defect propagating along the generatrix of the cone, the propagation direction does not change; and after the wall current of the hemispherical and semi-cylindrical defects flows into the defect, the tangential direction of the defect changes, and the energy is further lost.

From the distribution of electric field, magnetic field, and tube wall current at the defect, it can be seen that, with an increase in the defect volume, the more the direction of the tube wall current changes, the more the tube wall current leaks out of the defect, and the greater electric field and magnetic field in the rectangular waveguide. The larger the peak value of the tube wall current.

### 3.2. Microwave Detection Signal of TEO1 Mode Metal Surface Defect

When there is no defect on the metal surface, the microwave forms total reflection on the metal surface, and the reflection coefficient of the incident wave equal to the reflected wave is one. When a defect exists on the metal surface, the energy loss occurs at the defect, and the reflection coefficient is less than one. In practical engineering applications, the energy change of the microwave reflected wave at the defect is reflected by the return loss:(27)RL=−20lg|Γ|

#### 3.2.1. Rectangular Defect

The microwave excitation frequency was 5.2–6 GHz, and the propagation mode was TE01. The return loss of microwave reflected waves without defects, crack defects, and rectangular defects of different sizes is shown in Figure 11.

As shown in Figure 11, when the metal surface has no defects, the return loss is 0 dB; when the metal surface defect is a crack, the return loss is −0.11 dB; when the surface defect is a triangular prism, the return loss is −2.41 dB; and when the metal surface defect is a trapezoid, the return loss is −4.51 dB.

When there is no defect on the metal surface, microwave energy loss does not occur on the metal surface; with the appearance of metal surface cracks, microwave energy loss occurs near the metal surface defect. When the crack width was 0.3 mm, the defect detection frequency was 5.7 GHz; when the defect width was 10 mm wide, the defect detection frequency was 5.6 GHz. When the defect width decreased, the detection frequency increased. The energy distribution of the electric field, the magnetic field, and the wall current at the defect is related to the defect volume. It can be seen from Figure 12b,d that, the larger the defect volume, the greater energy distribution, the more energy loss, and the larger the return loss peak.

#### 3.2.2. Cylindrical Defect

The microwave excitation frequency is 5.2–6 GHz, the propagation mode is TE01 mode, and the return loss of the microwave reflected wave of the cone defect, hemisphere defect, and cylinder defect is shown in Figure 12.

As can be seen in Figure 12, when the metal surface defect is a cone, the return loss is −0.13 dB; when the metal surface defect is a hemisphere, the return loss is −0.25 dB; when the metal surface defect is a cylinder, the return loss is −0.27 dB; and when the defect is a half cylinder, the return loss is −5.612 dB.

When the metal surface defect is circular, the wall current propagates periodically on the inner wall of the rectangular waveguide, and the curved shape propagates along the inner wall of the rectangular waveguide. Therefore, for a circular metal surface defect, part of the wall current flows out along the defect, and the energy at the defect has fewer leaks, and microwaves are less capable of detecting surface defects on circular surfaces.

## 4. Experimental and Result Analysis

The metal surface defects were detected by using a vector network analyzer when the microwave propagation mode was the TE01 mode and the microwave excitation frequency was 5–6 GHz. The crack defects and the cylindrical defects of different sizes under the cylindrical coordinate system were detected. The laboratory intent is shown in Figure 13.

As shown in Figure 13, the vector network analyzer transmits and receives microwaves. The rectangular waveguide converted the TEM waves in the coaxial cable into TE01-mode microwaves propagating in the rectangular waveguide. The microwaves were reflected from the metal surface. The energy change was used to detect metal surface defects.

### 4.1. Microwave Inspection of TE01 Mode for Crack Defects with Different Depths

The crack width on the metal surface was 0.3 mm, the depth was 1–6 mm, the defect length was 5 cm, and the defect interval was 5 cm. Metal plates with different depths of crack defects are shown in Figure 14.

The vector network analyzer transmits and receives microwaves and analyzes the energy loss of microwave reflected waves at the defect. With an increase in defect depth, the return loss of microwave reflected waves under crack defects at different depths is shown in Figure 15.

As shown in Figure 15, as the depth of the defect increases, the detection frequency is 5.735 GHz. As the depth of the defect increases, the return loss values are −3.078, −4.72, −5.3, −5.45, −6.15, and −6.2 dB; the energy loss at the defect increases; and the return loss value of the microwave reflected wave at the defect increases.

### 4.2. TE01-Mode Microwave Detection of Different Types of Rectangular Defects

When the excitation frequency of a TE01-mode microwave is 5–6 GHz, there are triangular-prism defects (20 mm × 10 mm × 5 mm) and trapezoid defects (upper surface, 20 mm × 10 mm; lower surface, 20 mm × 4 mm; and depth, 5 mm) on the body (Figure 16).

The vector network analyzer detected a rectangular defect on the metal surface, and the return loss of the microwave reflected wave is shown in Figure 17.

Figure 17a shows that, when the metal surface is free of defects, the reflected wave has almost no energy loss, and the return loss is approximately 0. When the defect width is 10 mm, the microwave detection frequency is 5.6 GHz, the return loss value of the microwave detection signal of the triangular prism defect is −15.4 dB, and the return loss value of the microwave detection signal of the trapezoid defect is −25.93 dB. When the lower surface of the triangular pyramid is continuous, the return loss is small; when the lower surface of the trapezoid is discontinuous, there are many interfaces, and the return loss is large. These results are consistent with the simulation results.

### 4.3. Detection of Cylindrical Defects of Different Sizes by TE01-Mode Microwave

When the excitation frequency of the TE01-mode microwave was 5–6 GHz, the sizes of the defects to be tested were as follows: cylindrical defect (φ10 mm × 5 mm), cone defect (φ10 mm × 5 mm), hemispherical defect (φ10 mm), and semi-cylindrical defect (section is 20 mm × 5 mm × φ10 mm). The schematic diagram of the surface defect of the cylindrical defect metal is shown in Figure 18.

The step length of the rectangular waveguide is 1 cm. The microwave detection signal of the defect is shown in Figure 19 for when it passed through the metal surface defect, when the defect is in different positions in the rectangular waveguide.

It can be seen from Figure 19 that, when the microwave detection frequency is 5.6 GHz, the peak value of the return loss of the cone defect is −9.873 dB, the hemispherical defect is −14.172 dB, the cylindrical defect is −15.37 dB, and the semi-cylindrical defect is −29.9dB. Cylindrical, cone, and hemispherical upper surfaces are arc-shaped, resulting in less energy loss and return loss. When the wall current propagates along the inner wall of the defect, the tangential direction of the arc changes, and the wall current in the hemisphere changes. The direction of the tube wall and the inner wall of the semi-cylinder changes, and this hinders the current of the tube wall and causes a large energy loss. Cylindrical defects and hemispherical defects change in the direction of the inner wall of the defect, resulting in discontinuity, and the return loss is greater than that of the cone defect; the surface of the hemispherical defect is rectangular, and the direction of the inner wall of the defect changes, resulting in the largest return loss.

## 5. Conclusions

(1)In our study, we established a microwave detection model based on TE01-mode microwave in order to detect metal surface defects, established a relationship model between defect size and microwave reflection coefficient, and obtained a relationship model between microwave reflection coefficient and defect size;(2)Due to the existence of defects in the electric field, magnetic field, and tube wall current of the microwave propagation in the rectangular waveguide, the phase of the microwave propagation cycle is shifted, and the microwave energy is concentrated at the defect, resulting in energy loss and an increase in the return loss value.(3)The TE01-mode microwave can effectively detect 0.3 mm wide crack defects at 5.73 GHz, and the return loss value increases with the increase in defect depth; at 5.6 GHz, it can effectively detect 20 mm–wide trapezoid and triangular prism defects. The defect width increases, and the microwave detection frequency decreases; microwaves are more sensitive to crack defects with rectangular surfaces and more sensitive to defects with arc-shaped inner walls. The microwave has better detection ability for discontinuous surfaces.

In this paper, the metal surface defects were detected under experimental conditions. Next, it can be approximated as the actual detection conditions. The rectangular waveguide probe and the test piece contain lift-off values, and the rectangular waveguide probe and the test piece contain media of different materials. We detected metal surface defects, so as to realize the detection of large-scale metal surface defects such as storage tank bottom plates; from metal surface defect detection to small-diameter pipeline detection, it can serve urban pipeline networks, nuclear power pipelines, etc.

## Figures and Tables

**Figure 1 sensors-22-04848-f001:**
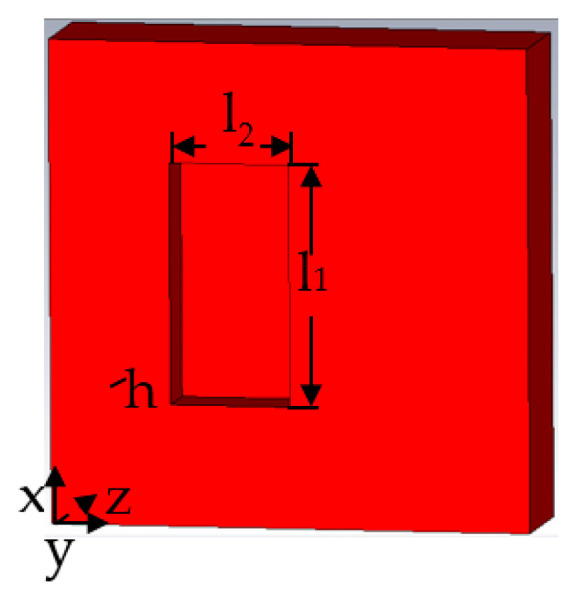
Three-dimensional model of rectangular defect.

**Figure 2 sensors-22-04848-f002:**
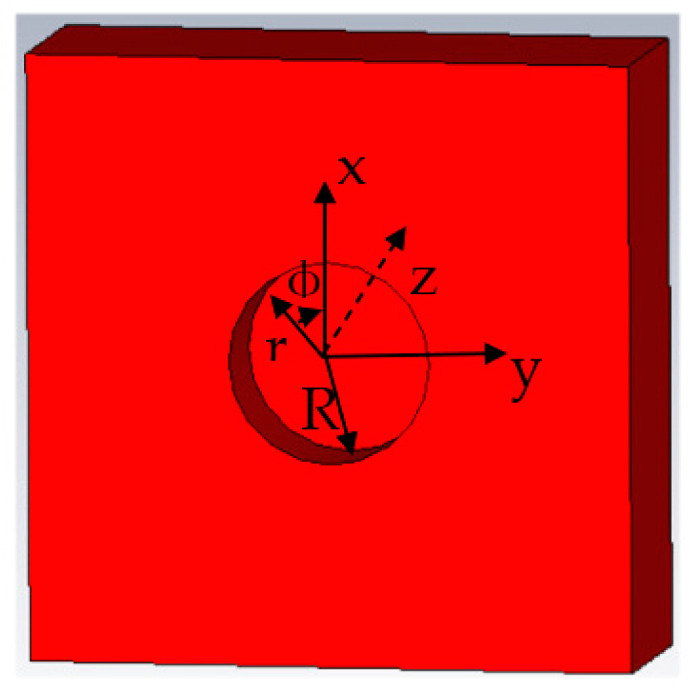
Three-dimensional model of cylinder defect.

**Figure 3 sensors-22-04848-f003:**
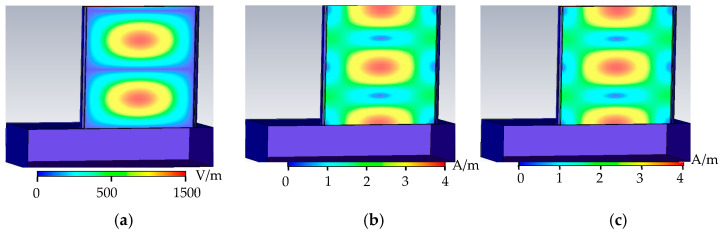
No-defect field distribution. (**a**) No defect electric field distribution. (**b**) No defect magnetic field distribution. (**c**) No defect wall current distribution.

**Figure 4 sensors-22-04848-f004:**
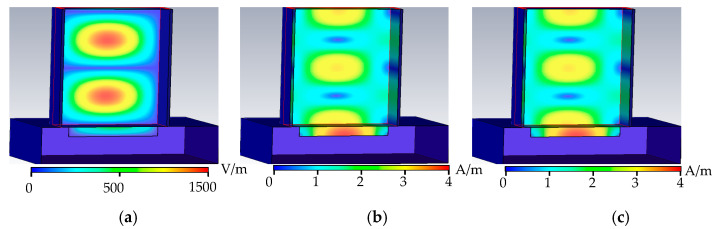
Crack defect field distribution. (**a**) Crack defect electric field distribution. (**b**) Crack defect magnetic field distribution. (**c**) Crack defect wall current distribution.

**Figure 5 sensors-22-04848-f005:**
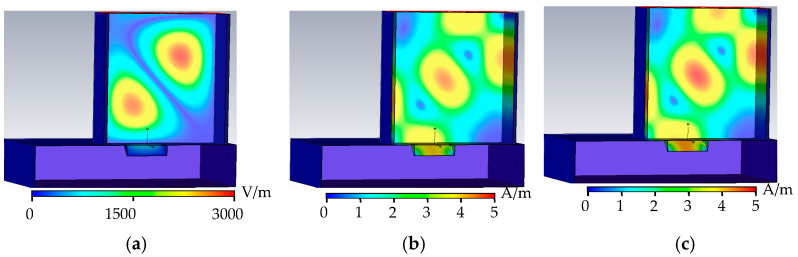
Triangular defect field distribution. (**a**) Triangular prism defect electric field distribution. (**b**) Triangular prism defect magnetic field distribution. (**c**) Triangular prism defect wall current distribution.

**Figure 6 sensors-22-04848-f006:**
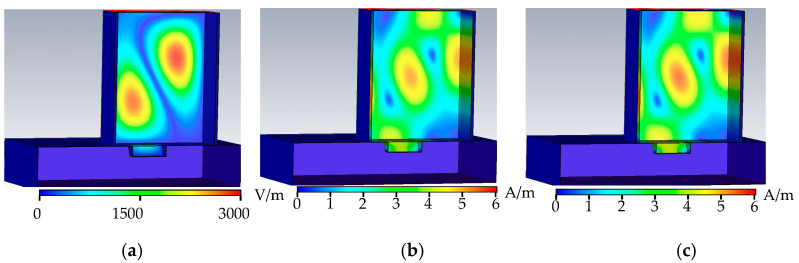
Triangular defect field distribution. (**a**) Trapezoid defect electric field distribution. (**b**) Trapezoid defect magnetic field distribution. (**c**) Trapezoid defect wall current distribution.

**Figure 7 sensors-22-04848-f007:**
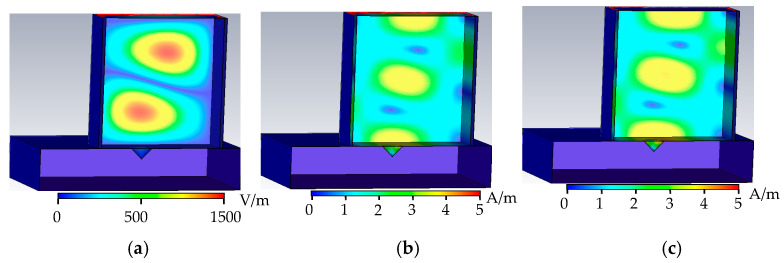
Cone defect field distribution. (**a**) Cone defect electric field distribution. (**b**) Cone defect magnetic field distribution. (**c**) Cone defect wall current distribution.

**Figure 8 sensors-22-04848-f008:**
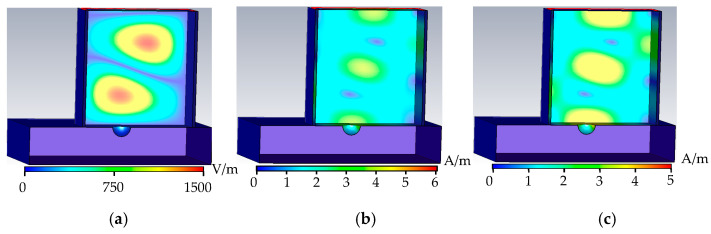
Hemispheric defect field distribution. (**a**) Hemispheric defectel ectric field distribution. (**b**) Hemispheric defect magnetic field distribution. (**c**) Hemispheric defect wall current distribution.

**Figure 9 sensors-22-04848-f009:**
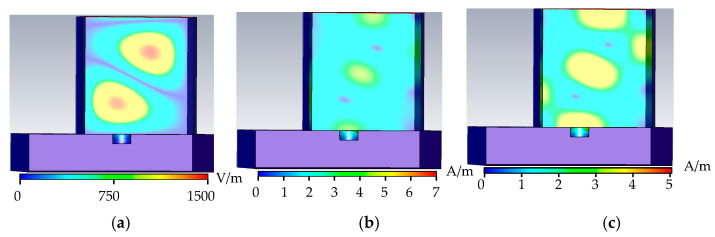
Cylinder defect field distribution. (**a**) Cylinder defect electric field distribution. (**b**) Cylinder defect magnetic field distribution. (**c**) Cylinder defect wall current distribution.

**Figure 10 sensors-22-04848-f010:**
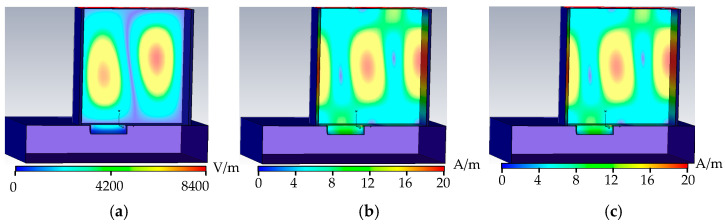
Semi-cylindrical defect field distribution. (**a**) Semi-cylindrical defect electric field distribution. (**b**) Semi-cylindrical defect magnetic field distribution. (**c**) Semi-cylindrical defect wall current distribution.

**Figure 11 sensors-22-04848-f011:**
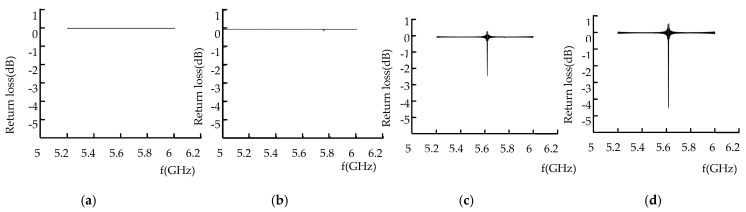
Return loss of rectangular defects with different sizes. (**a**) No defect. (**b**) Crack defect. (**c**) Triangular prism defect. (**d**) Trapezoid defect.

**Figure 12 sensors-22-04848-f012:**
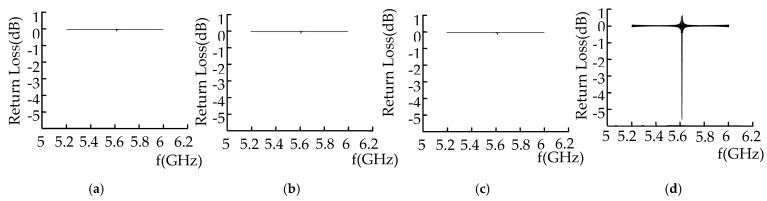
Return loss of different defects. (**a**) Cone defect. (**b**) Hemisphere defect. (**c**) Cylinder defect. (**d**) Half-cylinder defect.

**Figure 13 sensors-22-04848-f013:**
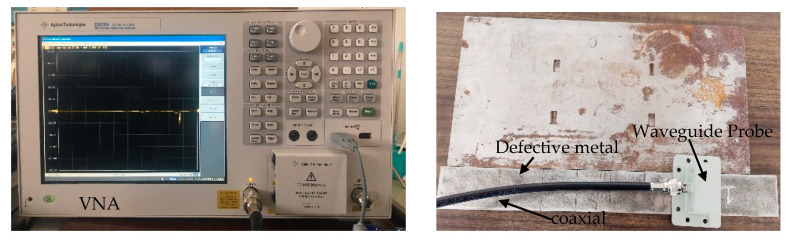
Schematic diagram of metal surface defect detection experiment.

**Figure 14 sensors-22-04848-f014:**
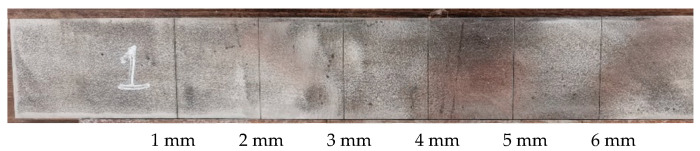
Metal plates with different depths of crack defects.

**Figure 15 sensors-22-04848-f015:**
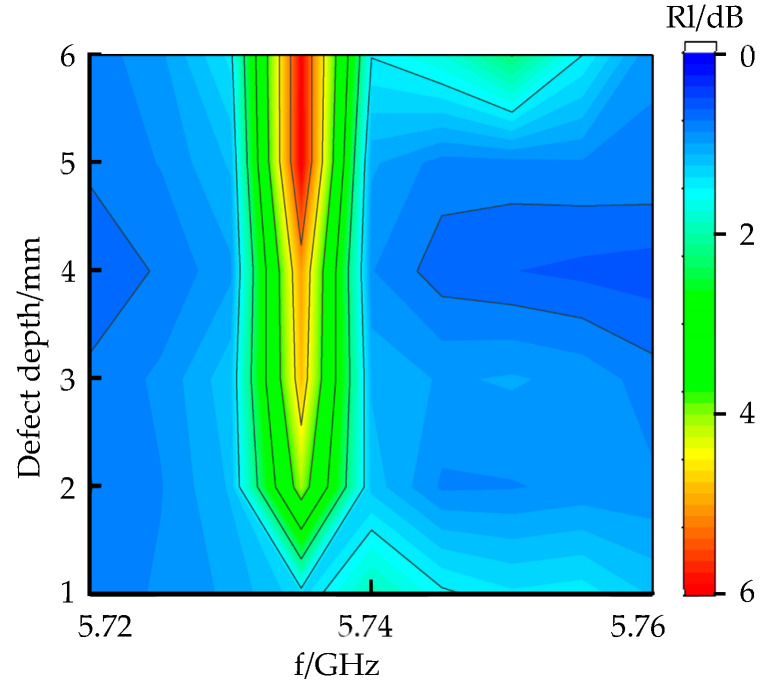
Crack defect signals at different depths.

**Figure 16 sensors-22-04848-f016:**
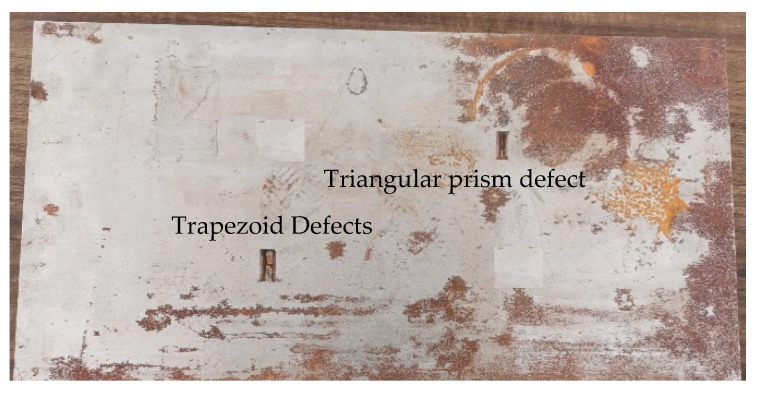
Schematic diagram of metal surface holding defects.

**Figure 17 sensors-22-04848-f017:**
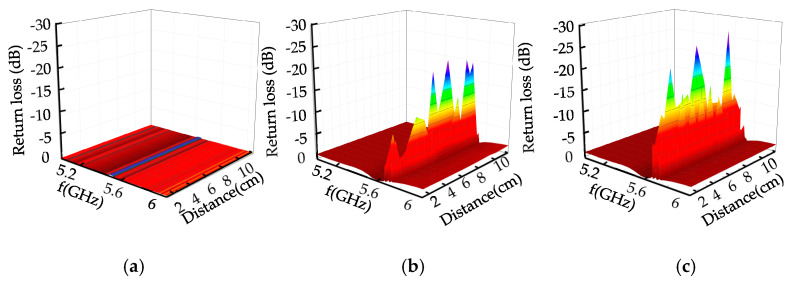
Microwave detection signals of different defects. (**a**) No defect. (**b**) Triangular prism defect. (**c**) Trapezoid defect.

**Figure 18 sensors-22-04848-f018:**
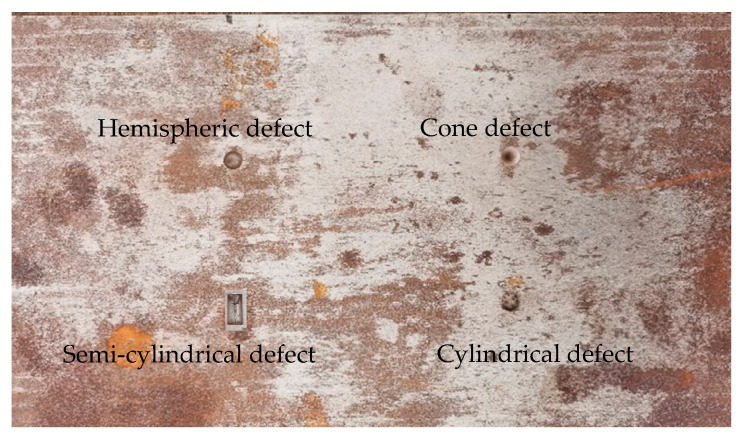
Schematic diagram of cylindrical defect on metal surface.

**Figure 19 sensors-22-04848-f019:**
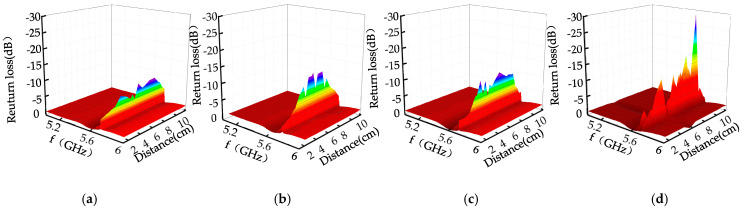
Microwave detection signals of different cylindrical defects. (**a**) Cone defect. (**b**) Hemispheric defect. (**c**) Cylinder defect. (**d**) Half-cylinder defect.

**Table 1 sensors-22-04848-t001:** Peak value of field distribution at rectangular defects.

	No Defect	Crack	Triangular Prism	Trapezoid
E (V/m)	1396.51	1401.97	2289.77	2594.5
H (A/m)	3.31	3.98	5.24	6.27
J (A/m)	3.31	3.97	4.86	5.86

**Table 2 sensors-22-04848-t002:** Electric field peak at different defects.

	Cone	Hemisphere	Cylindrical	Semi-Cylindrical
E (V/m)	1512.09	1553.4	1696.05	8389.74
H (A/m)	4.63	5.89	6.58	19.73
J (A/m)	4.12	4.43	4.67	19.67

## Data Availability

Not applicable.

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
