# Peer review of "Metal Surface Defect Detection Method Based on TE01 Mode Microwave"

_sensors, 2022, doi:10.3390/s22134848_

Round 1

Reviewer 1 Report

In this paper, a detection method for metal surface cracks and corrosion defects based on TE01 mode microwave is proposed. By establishing both simulation model and experimental platform, The TE01's ability to detect different types of metal surface defects is verified.

However, reviewer have some concerns are list as below:

1) The article is rough and there are many ambiguities. In page 6, line 193, “(50mm*0.3mm5mm)” is “(50mm*0.3mm *5mm)”? In page 6, line 223, “and the tube 222 wall current value is 3093.06V/m. 6.71A/m.” Whether 3093.06V/m is redundant? In line 226, “As shown in figure 7” should be placed by “figure 6”? In line 271, is 1420.37 “V/m”? In line 282, current, should “the ability to detect circular arc surface defects is weakened.”? In line 325, “It can be seen from figures c-figures e”, where is figures e? In line 389, is “4.3”?

2)In the “3.1 TE01 mode microwave defect detection model”, Table 1,Table 2 has information redundancy with the content expressed in the section. 3.1 should be more concise, the analysis should be more closely related to metal surface defect detection based on TE01 mode microwave.

3)Does the TE01 mode microwave only has good detection ability for the metal surface cracks and corrosion defects ?Are the modeling methods and conclusions in this paper adaptable to other types of defects such as inclusion and patch?

4)The label of the graph coordinate in this paper can be smaller to make the graph less crowded and more beautiful.

5) The English writing should be further improved and the article should be checked carefully.

Author Response

Dear Reviewer:

Thanks very much for taking your time to review this manuscript. I really appreaciate all your comments and suggestions. Please find my itemized in attachment and my corrections in the re-submitted files.

Thanks again!

Reviewer 2 Report

The work investigates the crack and corrosion detection in metals through a microwave technique. The analysis is based on both numerical simulation and experimental results. The work is overall of interest to the community and within the scope of the journal. However, important analysis and methodology issues need to be addressed. In particular:

1) The authors elaborate a numerical framework with for the defect detection. However, the novelty of the approach with respect to existing contributions is rather unclear. The novelty needs to be clearly highlighted.

2) No specific information is provided for the numerical analysis model employed. In particular, what are the elements used and how is convergence reassured? Relevant information needs to be added.

3) what is the range of validity of the explicated methodology in terms of both crack size and electric field magnitude? How is the limit of 0.3 mm stated in the abstract of the manuscript deduced?

4) What are the margins of the methodology applicability with respect to the  the depth, width and excitation frequency required? In Fig. 12, only half-cylinder defects appear to provide a substantial change in the return loss, with the other three cases to yield rather weak results. Clear statements with respect to applicability limitations need to be provided.

5) The TE01 mode needs to be described in more detail for completeness.

6) The literature review needs to be extended to account for different surface crack detection methods. In particular, different image-based methods have been presented and widely employed, such as https://doi.org/10.1016/j.aej.2017.01.020, https://doi.org/10.1016/j.ijmecsci.2021.106698.

6) The manuscript needs to be thoroughly proof-read and different language shortcomings and typos need to be corrected. Indicatively, in the abstract section, the 4rth line before the end should be corrected: ...was built. The absolute value...

Author Response

(The authors gave the same response as above.)

Round 2

Reviewer 1 Report

For improving the quality of the manuscript, I suggest the author carefully check format before official publication.In particular,  the pictures are not completely modified in uniform font size. The author must carefully correct his manuscript!

Reviewer 2 Report

The authors have addressed all issues raised and accordingly revised the manuscript. I suggest that the work is published in its current form.